# Percutaneous Transhepatic Electrohydraulic Lithotripsy for the Treatment of Difficult Bile Stones

**DOI:** 10.3390/jcm10071372

**Published:** 2021-03-29

**Authors:** Anna Maria Ierardi, Giovanni Maria Rodà, Letizia Di Meglio, Giuseppe Pellegrino, Paolo Cantù, Daniele Dondossola, Giorgio Rossi, Gianpaolo Carrafiello

**Affiliations:** 1Radiology Department, Fondation IRCCS Cà Granda Ospedale Maggiore Policlinico, Via Francesco Sforza, 35, 20122 Milan, Italy; gianpaolo.carrafiello@unimi.it; 2Postgraduation School in Radiodiagnostics, University of Milan, via Festa del Perdono, 20122 Milan, Italy; giovanni.roda@unimi.it (G.M.R.); letizia.dimeglio@unimi.it (L.D.M.); giuseppe.pellegrino@unimi.it (G.P.); 3Gastroenterology and Endoscopy Unit, Fondation IRCCS Cà Granda Ospedale Maggiore Policlinico, Department of Pathophysiology and Transplantation, University of Milan Via F. Sforza 35, 20122 Milan, Italy; paolo.cantu@policlinico.mi.it; 4General and Liver Transplant Surgery Unit, Fondation IRCCS Cà Granda, Ospedale Maggiore Policlinico, 20122 Milan, Italy; daniele.dondossola@unimi.it (D.D.); giorgio.rossi@policlinico.mi.it (G.R.); 5Department of Pathophysiology and Transplantation, University of Milan, 20122 Milan, Italy; 6Department of Health Sciences, University of Milan, 20122 Milan, Italy

**Keywords:** percutaneous lithotripsy, electrohydraulic lithotripsy, biliary stone, percutaneous choledoscopy, cholangiography

## Abstract

Objectives: To evaluate the efficacy and safety of percutaneous transhepatic lithotripsy (PTL) using an electro-hydraulic (EH) system for difficult bile stones. Methods: We retrospectively evaluated two patients with recurrent cholangitis, jaundice and fever for the presence of difficult bile stones, inaccessible by an endoscopic approach, treated with PTL. Both procedures were conducted using the same protocol, with two different accesses. The treatments were performed using a 10 Fr flexible choledoscopy SpyGlass DS^TM^ for visualization and an EH system for lithotripsy. Results: Technical success, clinical success and complications were evaluated. The two procedures were successfully concluded in both patients without any residual stones in the biliary tree. For both patients, a short follow-up period of six months was available, during which they remained asymptomatic. Neither major nor minor complications were registered. Conclusion: PTL was determined to be an effective and safe technique. This procedure allows a direct visualization of the stone, reducing fluoroscopy time and permitting a less invasive and less traumatic method for the percutaneous management of difficult bile stones. Advances in knowledge: The direct visualization, the high quality of the digital view, the adequate length of the device and the less traumatic approach of EH systems represent advantages compared with other available technologies.

## 1. Introduction

For patients for whom an endoscopic retrograde cholangio-pancreatography (ERCP) is not feasible—for conditions such as a stricture below the stone, impacted stones, difficult anatomic access to the papilla (such as in the presence of a duodenal diverticulum), Billroth-II anatomy or Roux-en-Y reconstruction—the percutaneous approach is the only feasible option [1].

In many of these situations, options include transhepatic stone removal, using a Dormia basket or pre-dilating the papilla with a balloon and clearing common bile duct (CBD) stones using occlusion balloon pushing. Laser lithotripsy and extracorporeal shock wave lithotripsy (ESWL) represent other therapeutic percutaneous options [2].

The recent introduction of the digital version of SOC (D-SOC; SpyGlassDS^TM^, Boston Scientific, Natick, MA, USA), available for clinical use since February 2015, has significantly improved image quality compared with the previous system, possibly increasing the diagnostic and therapeutic capabilities of cholangioscopy [3].

The electrohydraulic lithotripsy (EHL) technique uses a charge generator and a bipolar probe to create a spark to produce a vapor plasma. The vapor plasma becomes a cavitation bubble that oscillates around the tip of the probe, which leads to stone fragmentation by absorption of rebounding shockwaves from the vapor [4].

Patients that undergo percutaneous transhepatic lithotripsy (PTL) are generally referred to our service after discussion in a multidisciplinary meeting involving gastroenterologists, hepatobiliary surgeons and interventional radiologists [2].

We exhibit two cases with difficult biliary stones, inaccessible using an endoscopic approach, that successfully underwent percutaneous transhepatic cholangioscopy (PTC) with the SpyGlassDS^TM^ Direct visualization system and electrohydraulic lithotripsy (EHL).

## 2. Materials and Methods

We present retrospectively two patients with recurrent cholangitis, jaundice and fever, in which ERCP was not resolutive or was impossible.

Risks and benefits of the procedures were explained and informed consent was signed by both patients.

### 2.1. Patient 1

A 56-year-old man with COVID-19 infection, presented an obstructive pattern with high liver enzymes and positive blood cultures for Escherichia coli; therefore, a diagnosis of cholangitis was made. Magnetic resonance cholangiopancreatography (MRCP) showed a biliary stone in the right biliary duct at the confluence with the left biliary duct and a stricture of the CBD due to a post-operative complication after cholecystectomy. ERCP was performed but not resolutive, especially for intra-parenchymal calculi. The patient was first treated with bilateral percutaneous transhepatic biliary drainage (PTBD) in order to treat cholangitis (Figure 1). After the resolution of cholangitis and COVID-19 infection, PTL was performed (Figure 2A–E).

### 2.2. Patient 2

A 33-year-old man, with an history of left liver transplantation followed by a Roux-en-Y hepaticojejunostomy twenty years ago, presented with severe acute cholangitis and hyperbilirubinemia. MRCP revealed intrahepatic biliary ductal dilatation and 14 mm hypointense filling defect in the common hepatic duct, just proximal to the anastomosis and intra-parenchymal lithiasis (11 mm and 15 mm). PTC, pre-dilatation of the anastomosis with a balloon and an attempt to push stones using an occlusion balloon were performed. Only pre-anastomotic calculi were successfully pushed. Subsequently, PTL was performed (Figure 3A–F).

### 2.3. Technique and Procedure

Both procedures were conducted using the same protocol, with two different accesses. The PTL was performed using a 10 Fr flexible choledoscopy SpyGlass DS^TM^ (Boston Scientific). Through the 10 Fr shift (Terumo, Japan), the SpyGlass was advanced into the bile duct. Each patient was kept in a state of moderate sedation through intravenous administration of a combination of midazolam 0.07–0.08 mg/kg (Ipnovel15, Roche, Milan, Italy), propofol 0.5–2.0 mg/kg/h (Diprivan, AstraZeneca S.p.A., Caponago, Italy) and fentanyl 1–2 mg/kg (Fentanest; Pharmacia & Upjohn, Milan, Italy). Heart rate, electrocardiographic trace, oxygen saturation, respiratory frequency and blood pressure were continuously monitored throughout the procedure. During the entire procedure, patients were monitored by an anesthesiologist. Adequate antibiotic prophylaxis was achieved with 4.5 g of cefazolin sodium (Ancef, SmithKline Beecham Pharmaceuticals, Philadelphia, PA, USA), routinely administered for biliary procedures. All procedures were performed in the angiography suite under fluoroscopic guidance (Philips Azurion 7 B20/15, Philips, Best, The Netherlands). Fluoroscopy was used as needed for anatomic location and for intraprocedural cholangiography to show retained stones or stone-free status.

The SpyGlass Direct Visualization system is made by an irrigation tube, light source, access and delivery catheter (SpyScope), and optical probe (SpyGlass). The optical probe consists of two sections. The first one is a one-use SpyScope 10 Fr access and a delivery catheter with a 1.2 mm diameter working channel and dedicated irrigation channels. The device was introduced over 0.035 in guidewire. The catheter can deflect the tip in four routes with an inclination angle of at least 30°, allowing the operator to have a full view of the biliary tree. The second structure is a reusable SpyGlass Fiber Optic Probe that provides 6000 pixel images and a 70 degree field of view. This device can be reprocessed in order to allow multi-use of the latter (Figure 4) [5].

For fragmenting the stones, an electrohydraulic system is introduced from the dedicated canal of the choledoscopy and consists of a bipolar probe and a charge generator.

The generator (AUTOLITH, Northgate Technologies Inc, Elgin, IL, USA) is connected with a probe (Northgate Technologies Inc., Elgin, IL, USA) with a caliber of 1.9 F. Depending on the characteristics of the stone (impacted, type of stone), generator settings were adapted (with a minimum of 10 pulses per second and power output 60).

When a charge is transmitted across the electrodes at the tip of the probe, a spark is created. This induces expansion of the surrounding fluid, resulting in an oscillating shock wave of pressure that is typically adequate to fragment most stones. Saline solution irrigation is required to provide a medium for shock wave transmission, to assure visualization and to flush away debrids [6].

After stone fragmentation, a 10 mm occlusion balloon was used to push biliary debrids behind the first intestinal tract. Finally, a new choledoscopy and a cholangiography were performed to evaluate the persistence of bile stones/debrids. At the end of the procedure, a 10–12 F internal–external biliary drainage tube was left. Two to three days after PTL, a cholangiography was performed before removal of the biliary drainage.

## 3. Results

The time taken to complete the procedure was approximately one hour per patient. Technical success was defined as stone-free status of the biliary tract. Clinical success was defined as absence of clinical symptoms within the available follow-up. The two procedures were successfully concluded in both patients without any residual stones in the biliary tree. For both patients a short follow-up of 6 months was available, during which they remained asymptomatic. Neither major nor minor complications were registered [7].

## 4. Discussion

Compared to ERCP, PTL is invasive, time consuming and painful; nonetheless, it offers a less invasive alternative to open or laparoscopic surgeries, which are associated with greater complication risks [8,9].

The direct visualization of the SpyGlass and the high quality of the digital view (associated with the adequate length of the device studied for a percutaneous approach), the improved maneuverability, and the availability of independent irrigation channels to maintain a clear cholangioscopic field represent indisputable advantages compared with other existing systems [2,10].

When using the EHL system, the wall of the biliary tract does not suffer any trauma, such as bleeding or hyperemia, ensuring a safe procedure [11].

In very tight channels due to fibrosis, before introducing a spy scope, the use of a 12 F sheath may be considered, especially if bilioplasty is not satisfactory [12].

A recent multi-center study revealed that, in 28 patients, the effectiveness of percutaneous single-operator cholangioscopy using the SpyGlass for diagnostic and therapeutic indications may be considered as an alternative approach in clinical cases where gastrointestinal anatomy is altered or when ERCP has failed [13]. This system allows a direct visualization of the stones, reducing fluoroscopy time and permitting a less invasive and less traumatic method of the interventional radiology management of difficult bile stones that are not removable with other techniques.

To date, no data are available comparing PTL with EHL systems, lasers, or ESWL. More studies with longer follow-up periods are necessary to confirm advantages of PTL with EHL.

## Figures and Tables

**Figure 1 jcm-10-01372-f001:**
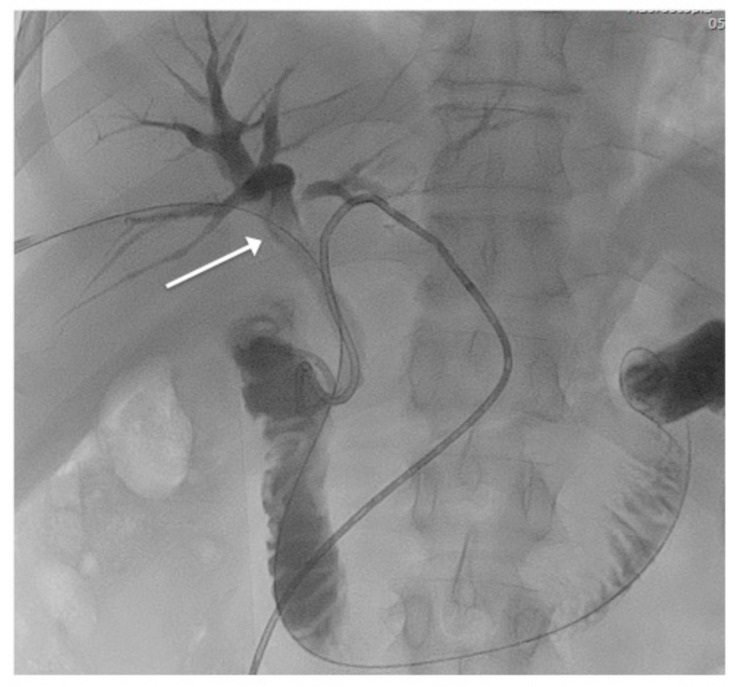
Percutaneous cholangiography confirmed the presence of a biliary stone (white arrow) in the right hepatic duct.

**Figure 2 jcm-10-01372-f002:**
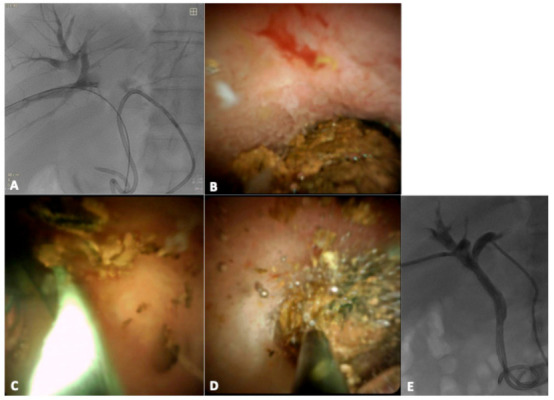
Percutaneous approach of the spyglass (**A**); intra-ductal visualization of the stone (**B**); intra-procedural lithotripsy (**C**) and wash with saline (**D**); cholangiography performed 3 days after percutaneous transhepatic lithotripsy (PTL) reveals absence of filling defects (**E**).

**Figure 3 jcm-10-01372-f003:**
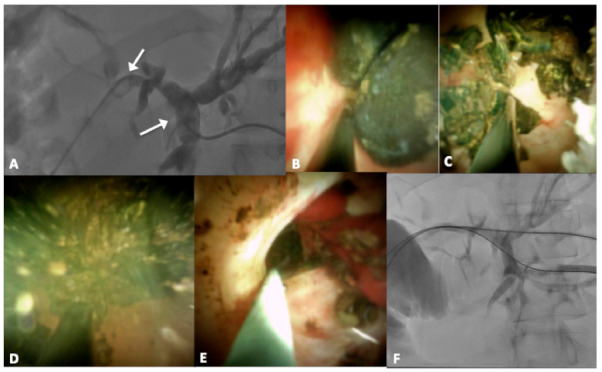
Percutaneous approach of the spyglass (**A**); intra-ductal visualization of the stones (**B**); intra-procedural lithotripsy (**C**), wash with saline (**D**) and residual biliary debris (**E**); cholangiography performed 3 days after PTL reveals absence of filling defects (**F**).

**Figure 4 jcm-10-01372-f004:**
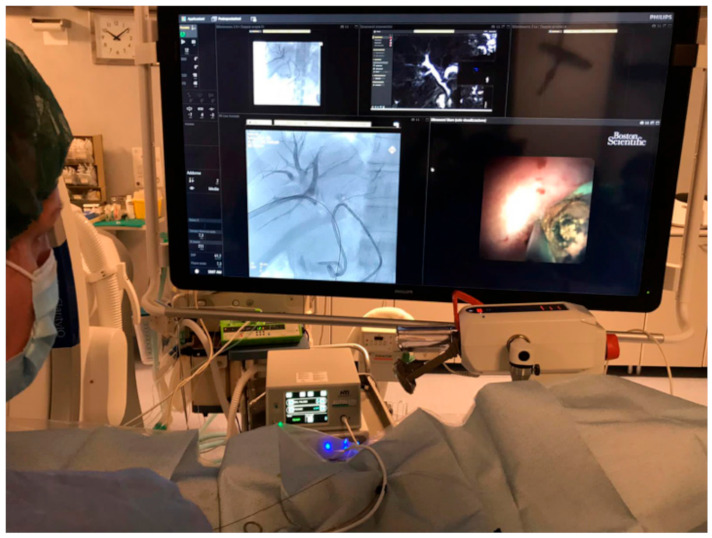
Monitors in front of the operators show fluoroscopic and choledocoscopic images.

## Data Availability

Data is available in the “References” section for each cited material.

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
