# Peer review of "Percutaneous Transhepatic Electrohydraulic Lithotripsy for the Treatment of Difficult Bile Stones"

_jcm, 2021, doi:10.3390/jcm10071372_

Round 1
Reviewer 1 Report
The current study was a retrospective case study of percutaneous transhepatic cholangioscopy (PTCS) using electrohydraulic lithotripsy (EHL) for bile duct stones. I have some comments as follows.
- The current study used Spyglass direct visualization system through percutaneous transhepatic route for the treatment bile duct stone. Several studies had already reported the effectiveness of Spyglass and EHL for removal of bile duct stones. What is the new finding in your study?
- The current study reported effectiveness of Spyglass legacy for removal of bile duct stones. However, Spyglass DS which has better imaging quality and irrigation system is now commercially available.
- Patient 1: The authors described that ERCP was not effective due to intra-parenchymal calculi. However, PTCS was nether effective in this situation. I assume that per-oral cholangioscopy could be effective and less invasive treatment for this patient.
- I could not confirm bile duct stricture in cholangiogram in Patient 1.
- The details of EHL generator and probe should be described.
- Hyperanemia is not usually used.
Author Response
- The direct visualization, the high quality of the digital view, the adequate length of the device and the less traumatic approach of EH system represent advantages compared with other available technologies. In the present study the utility of percutaneous device is described and our experience strengthens the already available literature.
- yes, we know
- ERCP was not resolutive for patient 1: papilla stricture was treated but intra-parenchymal calculi were not removed. That’s why cholangitis was not treated. Only percutaneous approach was successfully, as indicated in the text.
- Duct stricture is evident at confluence of right and left biliary duct (Fig 1); its resolution (Fig 2 E)
- The details of EHL generator and probe were provided, as you required.
- hyperanema was changed in hyperaemia
Reviewer 2 Report
The paper by Ierardi and colleagues is a "case report" of two cases of percutaneous transhepatic electrohydraulic lithotripsy performed with a device (a 10 Fr 23 flexible choledoscopy SpyGlass DSTM) interesting because this procedure allows a direct visualization of the stone, reducing fluoroscopy time and permitting a less invasive and less traumatic practice in the percutaneous management of difficult bile stones. Both cases were resolved successfully-
Other seven cases were reported in Pancreatology. 2018 Jul; 18 (5): 566-571. doi: 10.1016 / j.pan.2018.04.012. Epub 2018 Apr 30. In this series only one adverse event was noted wherein a patient had mild pancreatitis. This reference should be cited by the authors.
The paper is well written and the iconographic documentation is also clear.
Author Response
The paper “Pancreatology. 2018 Jul; 18 (5): 566-571. doi: 10.1016 / j.pan.2018.04.012. Epub 2018 Apr 30” was cited in the text, as you suggested.
Reviewer 3 Report
I enjoyed reading the submitted case presentation. Percutanous access to biliary system with the SpyGlass cholangioscopy device enables endoscopists to better detect biliary pathologies but also to improve minimal invasive therapies such as EHL in complicated biliary stones. It is a safe and effective procedure although larger case series are still rare. Therefore, the case presentation has a high educational impact.
I have some minor remarks to further improve the manuscript.
a) biliary access and of the Spy Scope:
- As far as I recognized, a guidewire has been placed over the PTC and the Spy Scope was introduced over the wire. This should be mentioned.
- How long has the PTC been performed before entering with the SpyScope? In case of a fibrotic "channel" around the PTC after a certain time the bilary Access with the SpyScope is easily to gain. In some cases he use of a 12F lock is mandatory (see also: Cholangioscopy-guided electrohydraulic lithotripsy of large bile duct stones through a percutaneous access device.Weigand K, et al. Endoscopy. 2018). This should be discussed.
- As far as I know, the Diameter of the SpyGlass catheter is 10.5 F.
b) EHL Probe:
- which probe was used for EHL? This needs to be specified.
-
Author Response
- Spy scope is introduced over a 0.035 inch guidewire: we introduced in the text this information, as you suggested.
In our experience blioplasty may be performed in fibrotic channel, before to introduce spy scope. We mentioned the paper published by Weigand K et al, as you suggested.
In very tight channels a 12 F sheat may be considered. We inserted this information in the text.
The external diameter of SpyGlass catheter is 10F. We used it with a 10F sheat (Terumo, Japan)
- The probe used was specified as you requested (probe (Northgate Technologies Inc, Elgin, IL, USA), with a caliber of 1.9 F.
Reviewer 4 Report
This is a brief Communication on the Percutaneous transhepatic electrohydraulic lithotripsy using the SpyGlass DSTM (Boston Scientific) cholangioscope. The authors presented 2 clinical cases of difficult bile stones demonstrating the efficacy and safety of the system.
Although the system has been on the market for a few years now, it is mainly used during ERCP and its application through a percutaneous access is limited. Some clarifications are necessary:
- Data from the literature suggest that, due to the small diameter of the cholangioscope, a dilatation of the percutaneous/biliary tract up to at least 11 French is necessary. In the manuscript the authors reported that the SpyGlass was advanced into the bile duct through a 10 Fr shift. Please verify if the diameter is correct and clarify this point.
- Please enhance the discussion with a brief review of the available literature on the use of the Spyglass for electrohydraulic lithotripsy with a percutaneous approach. A recent multicenter study has been published. (Percutaneous transhepatic cholangioscopy using a single-operator cholangioscope (pSOC), a retrospective, observational, multicenter study January 2021 Surgical Endoscopy DOI: 10.1007/s00464-020-08176), please discuss the results of this work in your manuscript.
- There are a number language/grammar mistakes throughout the manuscript and a professional English proof reading is mandatory .
.
Author Response
- We used a 10F sheat (Terumo, Japan) for percutaneous access
- The recent multicenter study wae reported in “Discussion” as you suggested.
- English revision was made by a native language speaker.
Round 2
Reviewer 1 Report
The authors modified the main documents in accordance with the comments.
This manuscript is a resubmission of an earlier submission. The following is a list of the peer review reports and author responses from that submission.